# The Evolving Landscape of Fungal Diagnostics, Current and Emerging Microbiological Approaches

**DOI:** 10.3390/jof7020127

**Published:** 2021-02-09

**Authors:** Zoe Freeman Weiss, Armando Leon, Sophia Koo

**Affiliations:** 1Brigham and Women’s Hospital, Division of Infectious Diseases, Boston, MA 02115, USA; Aleon4@bwh.harvard.edu (A.L.); skoo@bwh.harvard.edu (S.K.); 2Massachusetts General Hospital, Division of Infectious Diseases, Boston, MA 02115, USA

**Keywords:** fungal diagnostics, mycoses, invasive candidiasis, invasive mold infections, invasive aspergillosis, mucormycosis, transplant, immunocompromised host, non-culture diagnostics, culture independent

## Abstract

Invasive fungal infections are increasingly recognized in immunocompromised hosts. Current diagnostic techniques are limited by low sensitivity and prolonged turnaround times. We review emerging diagnostic technologies and platforms for diagnosing the clinically invasive disease caused by *Candida*, *Aspergillus*, and Mucorales.

## 1. Introduction

In recent years, the incidence of invasive fungal infections has increased in parallel with advances in chemotherapies, immunosuppression in solid organ and hematopoietic cell transplantation, and critical care technologies. The diagnosis of invasive fungal disease has traditionally relied on culture, direct microscopy, and histopathology. Conventional culture techniques are frequently insensitive, have prolonged turnaround times (TAT), and may require invasive sampling. An increase in the diversity of pathogenic species makes phenotypic identification challenging, particularly as the number of skilled clinical mycologists declines. Precise species identification is needed given the variability of antifungal drug susceptibility profiles even between closely related organisms. Thus, non-culture-based techniques have gained interest, particularly those with rapid turnaround times to allow for early clinical detection and decision making [1]. Here, we will review the existing diagnostic landscape, including some commercially available assays and platforms, and delve into promising emerging diagnostic techniques for the detection of *Candida* and the most common invasive mold infections, including *Aspergillus* and Mucorales.

## 2. Candidiasis

*Candida* is a commensal opportunistic organism that can become pathogenic in both immunocompetent and immunocompromised hosts. Though *Candida* is frequently responsible for superficial infections of the mucosal surfaces (such as in esophagitis or vaginitis), patients may present with more serious localized infections including pyelonephritis, endocarditis, or meningitis. Organ-specific disease is often due to hematogenous spread, particularly in the setting of foreign material (prosthetic valves, indwelling catheters, etc.) and impaired host immunity or anatomic abnormality. Immunocompromised patients are at risk for candidemia and deep-seated infection with visceral disease. Hepatosplenic candidiasis (also called chronic disseminated candidiasis) can result in intra-abdominal abscess or peritonitis and may or may not be associated with active candidemia (in some instances invading through the portal vasculature). Invasive candidiasis (IC) refers to active candidemia with or without deep-seated infection or deep-seated infection with or without active candidemia [2].

IC is the most common invasive fungal disease (IFD) in health care settings and mortality ranges between 40 and 60% [3]. Immunosuppressed patients or those with critical illness in ICU settings are at particularly high risk. More than 90% of invasive candidiasis (IC) is caused by 5 species—*Candida albicans*, *Candida tropicalis*, *Candida parapsilosis*, *Candida krusei*, and *Candida glabrata* [4], though there are over 30 species associated with human disease [5]. The distribution of these species varies geographically and there has been increasing detection of non-*albicans Candida* species, likely in relationship to selection pressure with the ubiquitous use of azoles and echinocandins. Recently, *Candida auris* has been identified as a major nosocomial pathogen [6]. Species-level identification is crucial given variations in drug susceptibility and virulence [7]. 

Blood cultures for detecting IC are positive in less than 50% of hematogenous disease and may be negative in cases of deep-seated infection without candidemia [8]. Cultures can take 1–5 days to show growth, and subsequent subculturing techniques can take an additional 24–72 h for identification. Delays in identification and subculture are in part due to the low number of circulating organisms, usually <1 colony forming unit (CFU) per mL [9], and slower multiplication rates compared to bacteria. Once blood cultures are positive, follow-up pathogen identification techniques are performed by microscopy, selective chromogenic isolation media, or biochemical/enzymatic testing on subcultures [10]. Automated platforms, such as the VITEK 2 system (bioMérieux, Durham, NC, USA), use fluorescence and biochemical features to perform species identification for a wide spectrum of yeasts and perform some antifungal susceptibility testing within 24 h [11]. 

Fungal cultures or histopathology from tissue or sterile body fluids are the gold standard for diagnosing IC in the absence of positive blood cultures [12]. Though 100% specific, a tissue diagnosis typically requires invasive sampling. Limited sensitivity, slow culturing and identification techniques limit early diagnosis [6,7]. Given the morbidity, mortality, and increasing drug resistance associated with IC, much attention has been paid to the earlier identification along with other non-culture-based methods [13].

### 2.1. MALDI-TOF MS 

Matrix-assisted laser desorption/ionization time of flight mass spectrometry (MALDI-TOF MS) platforms are widely used in clinical microbiology laboratories to identify bacteria and have increasingly been applied to yeast identification from positive cultures. A sample colony from a culture plate is placed onto a MALDI-TOF MS target plate and placed in an ionization chamber, generating a mass spectrum based on the mass-to-charge ratios of highly conserved ribosomal proteins, generating signature peaks that are then compared to reference samples within a database. This technique requires no prior knowledge of the organism and can be performed on multiple samples simultaneously, giving results in <10 min [14,15]. Accurate diagnosis requires adequate availability of reference strains in available, well-curated, and validated databases. Yeast requires some additional sample preparation as compared to bacteria. MALDI-TOF MS is available through multiple commercial platforms, and is now a standard method for identification of a wide spectrum of yeasts. MALDI-TOF MS has been shown to be more reliable in correctly identifying species of *Candida* (such as *C. auris*) than conventional techniques that may be prone to mislabeling closely related isolates, though this is dependent on the available spectra libraries [16]. Though identification is made quickly, this technology still depends on positive blood cultures and subsequent subculturing and isolation. Techniques for direct analysis from positive blood cultures have been developed to reduce time to identification but do not perform as well as subcultured samples [15]. Though MALDI-TOF MS has a high capital equipment cost, it is inexpensive to operate, has the potential for complete automation, and requires minimal technical expertise [14,17]. 

MALDI-TOF MS has also been used to determine antifungal susceptibilities. In 2009, Marinach et al. demonstrated that the minimal profile change concentration (MPCC), or minimum drug concentration needed to detect a change in MALDI-TOF spectra, and MIC were correlated and breakpoints could be established for antifungal susceptibility. This work evaluated the spectra changes in *Candida albicans* after exposure to varying concentrations of fluconazole [18]. The Bruker company developed a commercially available MALDI BioTyper Antibiotic Susceptibility Test Rapid Assay (MBT-ASTRA) (Bruker Daltonics, Bremen, Germany) which includes the detection of antifungal resistance by estimating growth in the presence of antifungal drugs (6 h) compared to a control setup without an antifungal. It is able to detect strains of *C. albicans* and *C. glabrata* that are resistant to caspofungin [19], and more recently has been applied to detect non-echinocandin-susceptible *C. auris* isolates [16]. MALDI-TOF susceptibility testing appears promising. However, agreement between conventional testing and MALDI-TOF may vary, thus conventional testing is still required [20]. Additional potential applications include typing to allow for identification of the geographic origins of specific strains to augment epidemiological tracing of *Candida* outbreaks (for example with *Candida auris*) [17]. 

Other promising spectroscopy techniques have been proven experimentally to detect *Candida*, though have yet to be applied in clinical laboratories and will be combined in a later section on spectroscopy techniques applied to yeasts and/or molds.

### 2.2. Antibodies and Antigen Biomarkers 

Serological testing is frequently used to detect evidence of invasive fungal infection in the absence of, or in conjunction with, culture data. Serological tests either detect antigens from an infecting agent or the antibodies formed in the host to those antigens. A number of latex agglutination tests, Western blot techniques, enzyme immunoassays, immunoprecipitation and immunodiffusion assays have been applied over the years to detect fungal antigens or antigen-antibody precipitates.

*Candida* polysaccharides and metabolites are important targets for serological testing. The 1,3-B-D-glucan (BDG) assay (Fungitell, East Falmouth, MA, USA), which was FDA cleared in 2004, is the most commonly used fungal antigen assay in clinical laboratories. BDG is a chromogenic quantitative enzyme-linked immunosorbent assay (EIA), designed to detect (1-3)-β-d-glucan polysaccharide cell wall component of *Candida* and other pathogenic fungi including *Aspergillus sp*. and *Pneumocystis jiroveci*. Sensitivity and specificity is widely variable in the literature at approximately 75–80% and 80%, respectively, for candidemia and ~65% and 75%, respectively, for intra-abdominal candidiasis [8]. BDG assays tend to be labor intensive and are generally performed in batch testing at reference laboratories [21]. Rapid TAT versions have been developed (see Table 1). The Wako-B-glucan test (Fujifilm Wako Chemicals, Richmond, VA, USA) is currently only commercially available in Europe. Though slightly less sensitive but more specific than Fungitell, its platform allows for multiple or single use, rather than batch testing, and has a TAT of ~120 min [22,23]. Furthermore, positive BDG assays in patients without IC can be seen with *Candida* or mold colonization, damage to the intestinal wall, hemodialysis, cellulose dressings, enteral nutrition, or mucositis, and should be only used with caution for screening or to guide pre-emptive antifungal therapy in high-risk immunocompromised patients [7].

Antibody testing for *Candida* has been employed to detect infection and has the theoretical potential to monitor disease response with titers. Commercially available antibody testing including the combined mannan (a polysaccharide target of the *Candida* cell wall [24]) and antimannan IgG tests (Platelia Candida Ag-Plus and Ab-Plus, Bio-Rad; Serion Mannan Kit, Serio GmbH, Bio-Rad, Marnes-la-Coquette, France) are used clinically in Europe but are not FDA cleared in the US. Though insensitive alone, their combined sensitivity and specificity is 83% and 86%, respectively, for candidemia and 40% and 25% for intra-abdominal infection [25]. The *C. albicans* germ tube antibody assays (CAGTA; Vircell Kit and VirClia IgG Monotest (Granada, Spain) detect antibodies to a *Candida* hyphal protein (Hwp1) using indirect immunofluorescence [28]. This assay has fewer supporting data, and reported sensitivities and specificities are 42–96% and 54–100%, respectively, for candidemia [8,30]. In one clinical study of ICU patients, use of the CAGTA assay was associated with lower mortality (presumably due to appropriate antifungal administration) and results were not affected by use of antifungal agents or *Candida* colonization, a limitation of other serological techniques [31,32]. Other antibody tests targeting *Candida*-specific enzymes, glycoproteins, secreted proteinases, and hyphal elements have been developed, though are not clinically used [33]. Antibody testing is less sensitive in immunocompromised patients and reliably distinguishing between infection and colonization remains a challenge.

Antigen-specific monoclonal and polyclonal antibodies have been proposed as a method of screening for *Candida* infection, but these antibodies tend to be highly specific for particular epitopes, limiting their scope of detection to isolated *Candida* strains. Cross-reactivity with other fungal species is also a major concern [13]. Recently, a bispecific monoclonal antibody targeted at β-glucan and MP65, an immunogenic mannoprotein secreted by *C. albicans* and other *Candida* species, (MP65/bglu mAb) has been developed in a murine model but has yet to be tested in a clinical setting [34].

A number of human biomarkers have been explored for diagnosing *Candida* infections and/or distinguishing between colonization and active infection. Interleukin-17, for example, is upregulated in responses to IC and is a potential marker to help distinguish between active infection and colonization, though its upregulation is not specific to IC and clinical application studies are needed [35].

New biomarkers for *Candida* and other fungal targets may be uncovered by application of investigational immunomic methods in research settings. Serological proteome analysis (SERPA), which uses high-resolution two-dimensional gel electrophoresis with Western blotting and mass spectrometry, has been applied to uncover new biomarkers. Such biomarkers can be used to detect IC, to characterize antibody/antigen patterns that can distinguish between colonization and infection, measure immune response, and identify targets for future vaccine development [36,37,38,39]. This technique involves profiling serological responses to peptides from cell surfaces. It favors the most abundant proteins and does not account for different stage and tissue-specific gene expression from cultured cells. Antigenic protein microarray technologies may overcome this limitation and have been applied to more precisely detect differences in IgG responses, profile host humoral responses during colonization and progression to candidemia, and detect antigens associated with drug resistance [36]. Recombinant complementary DNA expression libraries have been used to identify *C. albicans* genes expressed in host cells during active infection and identify virulence factors, and thus could be applied for identification of clinically relevant biomarkers [40]. Evolving antibody and antigen detection techniques can pave the way for the discovery of new clinically relevant biomarkers. Given the profound immunologic derangements in patients at highest risk for IFI, these techniques may be best applied in research settings, drug, and vaccine development.

Lateral flow assay platforms are an attractive option for rapid, affordable, diagnostic antigen testing in dipstick format. These require no technical expertise and are ideal for point of care (POC) applications, particularly in resource limited settings. Fluid (such as urine or blood) is applied to an absorbent surface and flows over a stripe of antibody-coated beads allowing for immunochromatographic antigen detection. Commercially available LFDs are already available for the detection of *Candida* antigens from cervical swabs [41]. LFDs have been developed to detect antibodies against *C. albicans* enolase for detection of invasive disease, though still needs clinical validation [42]. Dual path platform (DPP) devices (Chembio Diagnostic, Medford, NJ, USA), are enhanced immunochromatographic assays with improved sensitivity and multiplex capability. A DPP immunoassay has been developed to detect *Candida* from cultured cells but clinical application studies are still required [43].

In general, immunoassay techniques offer the possibility of rapid and even point-of-care diagnostic testing. However, due to concerns for cross-reactivity with other fungal species, low specificity, and reduced reliability in immunosuppressed hosts, these assays have limited clinical application at this time. Biomarkers that can measure host response or help distinguish between colonization and infection could potentially provide both diagnostic and prognostic information.

### 2.3. Nucleic Acid Detection 

Nucleic acid detection is a highly sensitive way of detecting the presence of *Candida* in clinical samples, providing genus- and species-level identification and, in some cases, detection of antifungal resistance genes. In-house and commercially available PCR assays vary significantly in their targets, including *Candida*-specific genes and highly conserved broad-range pan-fungal sequences, such as the 5.8S, 18S, 28S, ITS1 or ITS2 targets [29]. A 2011 metanalysis by Avni et al. included 54 studies (4694 patients) using various PCR techniques (such as real time, nested, or reverse-transcriptase-PCR) in single or multiplex (multiple simultaneous targets) formats and demonstrated a pooled sensitivity of 95% and specificity of 92% for patients with proven or probable invasive candidiasis vs. at-risk controls. Test performance was improved in those studies that used whole blood samples instead of serum or plasma, pan-fungal rRNA or P450 gene targets, primer-specific rather than multiplexed targets, and PCR detection limits of ≤10 CFU in vitro. This analysis also demonstrated that in patients with probable or possible IC, PCR had a higher positivity rate than blood cultures (85% vs. 38% in probable and 67% vs. 29% in possible, respectively). Turnaround times ranged from 4 to 12 h. Only 11 of 54 studies documented the presence of antifungal therapy before sampling, which is a notable limitation [29].

Multiplex syndromic panels that include a predefined number of pathogenic targets, often combining common bacterial, viral, or fungal pathogens, are becoming increasingly popular in clinical microbiology labs and can be applied to a number of different clinical samples (blood, CSF, tissue, etc.). A few commercially developed multiplex PCR platforms have been developed to detect *Candida* or other common yeast species directly from clinical specimens without requiring blood cultures (TAT 4–10 h), including Septifast (Roche Diagnostics, Manheim, Germany, Magicplex (Seegene, Seoul, South Korea) and VYOO^®^ (SIRS-lab, Jena, Germany) [24]. The SepsiTest. (Molzym, Bremen, Germany) is available in Europe and uses universal PCR with Sanger sequencing. Given the generally low prevalence of fungal pathogens in all patients with suspected blood stream infections, the positive predictive value of these assays as a screening strategy is limited.

Multiplex assays that can be run directly from positive blood cultures are becoming more broadly available for commercial use, although they requiring waiting for blood culture positivity. For example, The FilmArray BCID Panel (BioFire Diagnostics and bioMérieux, Salt Lake City, UT, USA), is an FDA-cleared, rapid (TAT = 1 h), 43 target (15 fungal, 5 major *Candida spp*) assay [44]. The ePlex BCID Panel-FP (Genmark, Carlsbad, CA, USA) is commercially available but not FDA-cleared, and detects up to 15 fungal organisms, including 11 *Candida spp.* TAT is approximately 1.5 h [45]. Multiplex assays have the advantage of being able to detect multiple organisms simultaneously, where cultures may only reveal one dominant pathogen in a polymicrobial infection. Cartridge-based platforms can be performed rapidly and with limited technical expertise. Further development of large-scale parallel amplification techniques with many targets are needed to detect a broader range of species [46]. Due to the limited number of targets, these high-cost systems need to be combined with other identification strategies to identify rare pathogens.

DNA microarray platforms have also been developed for *Candida* detection from positive blood cultures but are not commercially available. Gene-specific probes are attached to a solid substrate and labeled either using fluorescence or radioactive labels. Samples are added to the microarray, allowing for hybridization, and the subsequent fluorescent pattern is detected by microarray readers and amplified [13]. The Prove-it Sepsis platform (Mobidiag, Espoo, Finland) is a rapid broad-range PCR (3-h TAT) and microarray-based assay that detects 80 total bacterial and fungal targets from positive blood cultures with excellent sensitivity and specificity (99% and 98% for fungal targets) [47]. This assay has subsequently been discontinued, possibly due to its prolonged turnaround time compared to MALDI-TOF [48]. Application of this technology directly to clinical specimens, without requiring positive cultures, is an area of further potential development.

There are multiple downsides to nucleic acid detection techniques, including lack of standardization in DNA extraction, genetic targets, and clinical samples (whole blood vs. blood fractions). Traditional PCR techniques require strict temperature control, which limits their POC application. Isothermal methods of nucleic acid detection, such as loop mediated isothermal amplification (LAMP), NA sequence-based amplification (NASBA), and rolling circle amplification (RCA) [13], have been used to overcome this obstacle. Recently, LAMP techniques have been applied to POC assays for other pathogens, so may be an attractive option for application for rapid testing in limited resource settings in the future [49,50].

Primer-specific and multiplex platforms can identify either a single target or a pre-defined diagnostic spectrum of organisms, missing rare or emerging pathogens. DNA detection techniques, while highly sensitive, are unable to distinguish between pathogenic and commensal organisms [46]. False-positive results may occur due to exogenous sample contamination or similarities with human DNA. Assays that depend on positive blood cultures limit their POC application and those that require cell lysis preclude traditional susceptibility testing. Detection of known resistance mutations may be insensitive, as genes conferring resistance are often present in low copy numbers [51]. Interpretation of positive results from molecular testing in the setting of negative blood cultures is challenging and requires subsequent adjudication. Given an increasing reliance on syndromic multiplexed platforms, methodological standardization and validation in multicenter clinical studies are still needed to better define the clinical utility of these technologies.

Given the narrow spectrum of most primer-specific and multiplex assays (that identify just the most common pathogens), open-ended assays that can identify any species as long as their sequences exist within a reference database are attractive. Next-generation sequencing technologies can allow for whole-genome sequencing (WGS) to detect fungal pathogens without prior knowledge of the species in question. WGS could be applied directly to clinical samples. However, these complex techniques require expensive equipment and specialized laboratory skills, and require further clinical studies. The use of next-generation sequencing will be described in later sections [1,46].

### 2.4. T2 Candida, MR Technique

In 2014, the FDA cleared the use of the commercially available T2Candida assay, an automated qualitative nanodiagnostic instrument platform (T2Dx) (T2 Biosystems, Lexington, MA, USA) that readily identifies the five most common *Candida* species in whole blood samples without requiring blood culture, based on the DIRECT trial [52]. This platform amplifies *Candida* DNA using a thermostable polymerase and pan-*Candida* primers targeting ribosomal DNA intervening transcribed spacer (ITS) region 2. Once amplified, the product is then detected by amplicon-induced agglomeration of super magnetic particles and T2 Magnetic Resonance (T2MR) [21,52]. The T2Candida panel has a TAT of ~4 h and a reported overall sensitivity and specificity of 91.1% and 99.4%, respectively, for candidemia. Of note, the majority of *Candida* positive samples generating these estimates (250/256) were directly inoculated in the lab at varying concentrations rather than clinical samples from patients with candidemia [52]. When tested in only clinical samples (DIRECT2 trial), the sensitivity was similar (89% in 36 patients at the time of positive blood cultures) [53]. A 2019 meta-analysis of eight studies of T2MR found concordant sensitivity and specificity (91% and 94%, respectively), though there was significant heterogeneity among studies. [26]. The sensitivity of this test in non-candidemic patients with intra-abdominal candidiasis was much lower at 33% with preserved specificity in one study of 48 patients [27]. 

In patients with candidemia, T2MR was noted to frequently remain positive after treatment was initiated, potentially suggesting greater sensitivity than blood cultures in the subset of patients who had received antifungal therapy prior to testing (45% vs. 24%) [53]. The follow up STAMP trial validated this observation, showing that the T2MR assay outperformed blood cultures for monitoring the clearance of candidemia [54]. Diagnostic performance might be improved, particularly in non-candidemic patients, by combining T2MR with BDG [27,55]. Recent outcome studies have suggested that the use of T2MR has been associated with earlier antifungal discontinuation [56] but further clinical endpoints are still needed. Due to cellular lysis during isolation, T2MR techniques do not allow for drug resistance and susceptibility testing, a notable limitation. High costs, the need for specialized equipment, and the limited diagnostic spectrum of this technique are additional barriers to broader use. Further studies are needed to evaluate the role of serial sampling, to better characterize the effect of antifungal therapy, and to understand how to interpret and apply discrepant results when blood cultures are negative but the T2 results are positive [21].

### 2.5. Immunohistochemistry 

Fluorescence in situ hybridization (FISH) uses nucleic acid probes to identify pathogen-specific ribosomal RNA sequences. There are commercially available peptide nucleic acid fluorescent in situ hybridization assays (PNA-FISH) that come in multiprobe systems and have high sensitivity and specificity for identifying the five most common *Candida spp* from positive cultures after a period of fungal growth (19–75 h) [57]. Clinical application studies for the FDA-approved Yeast Traffic Light PNA-FISH (OpGen, Woburn, MA USA) demonstrated reduced empiric echinocandin use and cost savings. TAT is 30–90 min. The newer QuickFish (Opgen, Woburn, MA, USA) can identify bacteria or yeasts from positive blood cultures in less than 20 min [58]. Further development of FISH techniques applied directly to blood samples is needed to obviate the need for positive blood cultures [57]. These platforms do not require significant capital equipment costs and have the potential for high throughput testing. 

The Accelerate Pheno system is a commercially available and the FDA approved the ID and rapid phenotypic AST platform that detects a wide range of bacterial targets and 2 species of Candida (*C. albicans*, *C. glabrata*) from positive blood cultures using FISH for identification and morphokinetic cellular analysis for susceptibility. Subculture is not required; however, the panel for detecting yeast species is limited [59]. 

### 2.6. Microfluidic Devices 

Lab-on-a-chip devices that use microfluidic principles have been developed to detect fungi in the blood stream rapidly and represent a promising technology for rapid point-of-care testing. Microfluidic techniques separate and concentrate fungal cells directly from blood samples without requiring positive cultures. A number of techniques have been described, including the use of inertial forces [60], ligand-coated beads to capture and isolate cells with magnetic force [61], and fungal-specific antibody-coated channels [62,63]. The microfluidic techniques can be combined with nucleic amplification techniques or other methods of detection, such as mass spectroscopy [13]. Platforms that do not lyse the fungal cell wall can allow for subsequent susceptibility testing [63]. Asghar et al. developed the first immune-based microfluidic device that detected *Candida* from spiked saline and human whole blood [63]. Further development for research and clinical application is still needed. 

### 2.7. Conclusion

Diagnosis of candidemia has traditionally relied on culture-based phenotypic, biochemical/enzymatic, and immunologic approaches. Advances in existing technologies, including MALDI-TOF and multiplex PCR platforms, have allowed for accurate species diagnosis from cultures. Given the time delay in waiting for positive cultures, which is often on the order of a few days, culture-independent diagnostics are of increasing importance. Antibody/antigen-based assays, immunohistochemistry, nucleic acid detection techniques, and T2MR are increasingly being used in clinical settings. POC platforms such as LFDs and lab-on-a-chip devices have attractive features and may, in the future, become options for rapid testing for IC. 

## 3. Diagnosis of Invasive Mold Infections 

Invasive mold infections (IMI) are a major cause of morbidity and mortality in immunocompromised patients. *Aspergillus* is the most common opportunistic mold, though Mucorales, *Fusarium*, *Scedosporium/Lomentospora/Pseudallescheria*, and *Paecilomyces/Purpureocillium* are increasingly seen in clinical settings, especially in patients receiving mold-active antifungal prophylaxis [64]. In the absence of microbiologic data, the diagnosis is typically made clinically, with consideration of host factors (e.g., solid organ or hematopoietic cell transplantation (HCT), prolonged steroid use or exposure to immunosuppressants that impair T-lymphocyte function), radiographic appearance, and mycological evidence (e.g., antigen detection). The EORTC-MSG consensus guidelines outline specific criteria for categorizing IMI into possible, probable, or proven disease. However, the classification system is designed for research purposes rather than directing clinical management [12]. In practice, patients are frequently treated with empiric antifungal therapies without a definitive diagnosis, which can result in unnecessary exposure to toxic and costly medications or inadequate treatment in the setting of drug resistance.

Making a definitive diagnosis of invasive mold infection requires positive culture from sterile material or histopathology demonstrating hyphal invasion [12]. Biopsy may be infeasible or unsafe due to the location of infection, or risk of bleeding in patients with thrombocytopenia. Cultures are often insensitive, particularly early in the disease course, and slow growth can delay diagnoses for days to weeks. Colony morphology and microscopic identification from culture and histopathology are laborious, require skilled mycologists, and are not practical for the identification of rare species [14]. 

Ribosomal sequencing for tissue diagnosis is frequently performed on clinical samples, though sensitivity is variable. Immunohistochemistry may be applied to tissue samples to help distinguish between *Aspergillus* and Mucorales based on their morphological features, without waiting for positive cultures [65]. Proteomic techniques such as MALDI-TOF are increasingly being adapted to rapidly make a species diagnosis from prepared cultures in culture-positive cases, which comprise the minority of invasive mold infections overall [66,67]. In the absence of culture data, fungal markers like BDG and galactomannan, while non-invasive, are limited by poor sensitivity and specificity and do not apply to all species of mold. 

Development and clinical validation of diagnostic tests to detect mold infections is often hindered by the relatively low frequency of cases seen at any single institution and the need for an array of different specimens (blood, serum, plasma, BAL fluid, urine, etc.) for validation [30]. Unlike bacteria, working with mold specimens can be technically challenging. They may be unevenly distributed in samples, exist in different forms at various stage of growth, and have hardy cell wall structures that can make nucleic acid extraction difficult [68]. Debate over nomenclature of mold phylogenies also hinders streamlined standardization. In addition to the need for rapid techniques to identify mold species from cultures, rapid non-invasive diagnostic tests are needed to detect IMI and ideally, antifungal resistance, to better guide antifungal therapy.

Here we will discuss the current (see Table 2) and emerging diagnostic techniques (see Table 3) applied to diagnosing invasive mold infections, including pan-fungal diagnostic strategies as well as specific techniques emerging for *Aspergillus*, Mucorales, and other less common molds.

### 3.1. Ribosomal Sequencing

Clinical specimens, including tissue, blood, BAL fluid, and CSF, can be sent to reference laboratories for ribosomal sequencing for species-level identification. Amplification of highly conserved regions of fungal ribosomal RNA including the internal transcribed spacers 1 and 2 (ITS1 and ITS2) and the D1/D2 regions of the 28S rRNA gene, followed by sequencing, can allow for identification of a broad array of fungal species including rare organisms [69]. The sensitivity and specificity of pan-fungal sequencing techniques vary widely depending on the method of DNA extraction, the type and preparation of clinical sample, and whether hyphal forms are visible on histopathology [70,71,72]. Formalin fixation can reduce assay sensitivity due to DNA degradation (e.g., from 100% to 90% in one study [70]). Samples collected from non-sterile sites may reveal non-pathogenic commensal organisms of uncertain clinical significance [73]. The method of sampling is also important, where open resection provides a better diagnostic yield than FNA or core needle biopsy [70]. Most importantly, sensitivity is highest (>90%) [70] if fungal forms are visualized on histopathology [70,74]. Because of this, the EORTC/MSG recommends sequencing from tissue samples only if fungal elements are present [74]. Use of pan-fungal PCR in samples where fungal forms are not visualized on a clinical sample may help to augment a diagnosis if positive, but must fit clinically and cannot be used to exclude disease, particularly in patients receiving anti-mold therapies where assay sensitivity may be further limited [69]. Lack of technical standardization has been addressed by recent attempts to protocolize PCR methods from tissue samples (https://fpcri.eu/, accessed on 1 December 2020).

### 3.2. Next-Generation Sequencing 

Next-generation sequencing (NGS), also called high-throughput/massively parallel sequencing, is a non-culture-based technique that allows for the application of both targeted and whole-genome sequencing (WGS). There are a number of available sequencing technologies and data analytic software packages [75]. All human and microbial DNA is extracted from clinical samples (including blood, CSF, BAL fluid, etc.), without a priori knowledge of a particular target. Sequencing is performed using a “shotgun approach,” human DNA is removed, and results are compared to existing nucleotide sequences from in pre-formed databases. This can allow for identification of esoteric species, as well as potential resistance mutations, provided the sequences exist in a reference database. TAT is typically 12–24 h once it has been received by the reference laboratory [76]. 

NGS techniques enable evolutionary tracing and were used to identify outbreaks of various fungal infections including cases of *Exserohilum rostratum* meningitis related to contaminated injections [77], *Sarocladium kiliense* bloodstream infections from contaminated anti-emetic medication [78], and invasive wound mucormycosis [79]. There are a few commercially available NGS platforms that detect cell-free DNA (mcf-DNA-seq) from plasma (Karius, Redwood, CA, USA), DNA and RNA from cerebrospinal fluid (University of California, USA) and respiratory secretions (IDbyDNA, Salt Lake City, UT, USA) to diagnose fungal pathogens (in addition to bacterial and viral pathogens) [75]. A small number of studies have been performed to address the diagnostic utility of this technique. 

Small studies have shown good concordance of NGS with biopsy proven IFI [80,81]. One retrospective cohort study of 82 Karius tests ordered for suspected infection (representing 66 patents) in a varied patient population reported a positive impact only 6/82 cases (7.3%), a negative impact in 3 cases (3.7%), no impact in 71 cases (86.6%), and was indeterminate in 2 (2.4%) [82]. Thus far, the majority of studies are limited by small sample size, the inclusion of patients with suspected IFI from varied anatomic sites (e.g., lung, skin, sinuses) and lack of a control group. In one recent retrospective case–control study of 114 HCT recipients, overall sensitivity of NGS for proven/probable IFD was 51%, 31% for Aspergillus and 79% for non-Aspergillus IFD. There were two proven IFD cases where Karius testing was positive and both serum and BAL galactomannans were negative, and only one case of Aspergillus detected in a patient with possible IFD. Only one patient with possible IFD had a pathogen detected. Specificity was reported at 95%. Thus, this diagnostic modality has low to moderate sensitivity but potentially high specificity in patients with proven or probably pulmonary IFD. Sensitivity was improved when combined with GM or when samples were taken within 3 days of a clinical diagnosis. This assay is potentially useful as an adjunctive diagnostic technique in patients with a very high likelihood of proven or probable pulmonary IFD, with slightly better performance in non-Aspergillus IFD, although the assay cost and need for specimen shipping to a central laboratory may be barriers to adoption [83]. 

There are significant limitations to unbiased NGS techniques. Commercially available assays are typically expensive and though results are typically available within 24 h, TATs may be delayed due to the need to ship samples to specialized laboratories. Capital equipment costs, the need for highly trained laboratory staff, and comprehensive reference databases have been major limitations to adoption of this technique outside highly specialized reference laboratories. Positive results may represent contamination or identify non-pathogenic commensal organisms. Validation for rare species due to need for positive controls can be challenging [69,76]. Overall, low sensitivity precludes the use of NGS for stand-alone testing or to rule out infection. NGS may be useful as an adjunctive test in cases where invasive biopsy is contraindicated [84]. With decreasing costs and expanding databases, this technique is likely to be implemented more broadly [69,76]. The clinical application and stewardship of NGS sequencing technologies for diagnosing infection still requires clarification. 

### 3.3. MALDI-TOF MS 

Species-level identification of molds grown in culture is frequently desired in order to guide antifungal choice. MALDI-TOF MS has the advantage of being able to identify a wide spectrum of species from commercial and in-house databases [85]. Application of MALDI-TOF MS to filamentous fungi has evolved over the past ten years, but time-consuming sample preparation techniques, which can vary between manufacturers, and limitations of spectral databases and available isolate challenge sets have delayed its widespread use. The mechanism of culturing mold isolates and the stage of fungal growth may impact the identification, as different levels of mycelia and spores are present in liquid versus solid media, which have different proteomic fingerprints [14,66]. A number of studies have reported identification rates of filamentous fungi ranging between 15% and 97%, depending on the platform and database used. There was a notable trend towards using lower species-level cut offs, a log(score) that refers to the level of similarity between an unknown tested specimen and reference sample [86], to achieve higher detection rates with only marginal increases in false positivity. A score of ≥1.7 rather than the manufacturer recommended cut off of ≥2 has been widely adopted for fungal isolates [14,86] There are a number of commercially available platforms and significant differences between their curated databases, including the range of species included and the nomenclature used for species identification, which can make generalization somewhat challenging [14]. 

MALDI-TOF MS is a reasonable alternative to conventional microbiological and molecular methods for species identification from positive cultures, though lack of standardized processing techniques and incomplete database spectra are still limiting factors. Additional molecular diagnostic techniques are needed in cases that cannot be identified. 

### 3.4. Other Spectroscopy Techniques 

A variety of other spectroscopy techniques have been applied to fungal diagnostics in the research setting and have the potential for both accurate yeast and mold identification, particularly for use on direct clinical samples (rather than subcultured isolates). 

Rapid evaporative ionization mass spectrometry (REIMS) performs MS analysis of the metabolites produced by heating up cells to a gas-phase and identifies microbes based on their lipid content. This technique demonstrated 98–100% accuracy in identifying *Candida* isolates [87,88]. REIMS has been coupled with electrosurgery and is used for immediate intraoperative tissue identification for malignant tumors [89]. Based on these proof of concept studies, its application in fungal identification is a potential area of future exploration. For example, intraoperative identification of invasive mold infections could allow for immediate therapeutic decisions.

Fourier transform infrared (FT-IR) spectroscopy and Raman spectroscopy (RS) [46] use vibrational spectroscopy-based biochemical profiling to detect pathogen species at extremely high resolution. Raman spectroscopy has extremely high specificity for pathogen detection, though enhancement techniques, such as surface-enhanced Raman spectroscopy (SERS) are required to achieve good sensitivity. SERS uses metallic nanostructures to enhance scattering and could be a potentially useful tool for sensitive biomarker detection and can be applied directly to clinical specimens. SERS has been coupled with PCR techniques (e.g., used in the commercially available RenDx Fungiplex^®^, Renishaw Diagnostics assay, Glasgow, UK) for the detection of *Candida* and *Aspergillus*, but its clinical utility is not yet well defined [90,91] Interference-enhanced Raman spectroscopy is a slightly less sensitive but more economically feasible technique (easily fabricated substrates and long-term stability of substrates). It has been applied to the diagnosis of aspergillosis via detection of TAFC fungal siderophore (see Section 4.4) from urine samples with a <3 h TAT [92].

PCR coupled with electrospray-ionization mass spectrometry (PCR/ESI-MS) is a promising technique for identification of species-specific sequences in specimens containing visible hyphae and has been successfully applied to the detection of Mucorales in one study, though could theoretically be applied to other species. TAT is about 6h, although implementation is limited by the significant expense and limited availability of this technique [93].

## 4. Aspergillus

*Aspergillus* is a ubiquitous airborne environmental mold that can cause invasive aspergillosis (IA) in immunosuppressed patients. Patients at risk for *Aspergillus* infection include those with prolonged neutropenia, solid organ transplantation, HCT, or exposure to steroids or T-lymphocyte immunosuppressants. For pulmonary aspergillosis, the most common manifestation of *Aspergillus* infection, respiratory cultures, bronchoalveolar lavage (BAL) and lung biopsy are typically performed to obtain cultures. BAL yield is reduced in patients on antifungal therapy or those that have peripheral lesions. Culture is insensitive, can reveal colonization rather than infection, and can sometimes take days to weeks to yield a result [94]. Identification is often further delayed due to the need for sporulation in order to make a phenotypic identification. Fungal elements may be seen with calcofluor white staining. On histopathology, Gomori methenamine silver or periodic acid-Schiff staining are frequently used, though these stains are not specific for *Aspergillus* [95].

### 4.1. Serologies and Biomarkers

Clinicians currently heavily rely on serum fungal markers including the BDG and galactomannan (GM) to help establish or provide supporting evidence for the diagnosis of invasive aspergillosis in the absence of culture data. BDG and galactomannan are often ordered in parallel in patients with suspected aspergillosis. A wide range of reported sensitivities and specificities have been described in the literature (See Table 2).

BDG, as described previously, has a relatively high negative predictive value for excluding IFI, but is neither sensitive nor specific for *Aspergillus spp.* The galactomannan Platelia *Aspergillus* EIA/Ag (Bio-Rad, Redmond, WA, USA) assay is a monoclonal Ab immunoassay that detects branched β-1,5-linked galactofuranose side chains of the α-linked mannosyl backbone of the large GM polysaccharide, a component of the *Aspergillus* cell wall [96]. At an optical density index (ODI) of 0.5 the pooled sensitivity and specificity for proven or probable IA in serum samples is approximately 78% and 85%, respectively. Sensitivity decreases and specificity increases at higher ODIs. The GM assay is fairly specific for *Aspergillus*, can be used in serial monitoring to assess treatment response [97,98], and is FDA-cleared for detection in serum and BAL fluid, though it can be found in other bodily fluids (CSF, pleural fluid). Cross-reactivity in patients with histoplasmosis, fusariosis, and talaromycosis can occur [99]. The diagnostic performance of GM is dependent on the optical density cut off used to interpret positivity, the net state of immunosuppression of the host (higher sensitivity in neutropenic patients), and the presence of antifungal therapy [95]. Though this can be performed in hours, batch testing and use of reference laboratories can delay TAT to days, which can limit its use in early clinical decision making. There are a number of alternative biomarker detection kits that have come onto the market but still need proper validation.

Antibody testing for *Aspergillus* is available, though its application for the diagnosis of IPA is limited, given the weak and variable immune response elicited in neutropenic or immunosuppressed patients. Antibody testing is available for patients with suspected allergic or chronic cavitary aspergillosis, but will not be reviewed here [100,101].

### 4.2. Lateral Flow Devices 

Immunochromatographic lateral flow assays for IPA have been developed for POC (TAT ~ 15–30 min), rapid testing. The AspLFD (OLM Diagnostics, Newcastle upon Tyne, UK,) and the *Aspergillus* galactomannan LFA (IMMY) are two such assays, currently available in Europe. The LFD assay uses a JF5 antibody to detect a mannoprotein antigen released in serum and BAL during active fungal growth [102,103]. Like the GM-EIA, the LFA assay targets galactomannan but uses two mABs which may provide greater sensitivity [104]. It has demonstrated good qualitative agreement with GM-EIA [105]. Both assays show better performance in BAL fluid than serum, and among hematology patients as compared to other patient subgroups [104]. In patients with hematological malignancies, sensitivity and specificity of the LFD and LFA from BAL were 78%–89% and 88%–100%, respectively [106]. When applied to non-neutropenic/hematologic malignancy patients, sensitivity and specify were 58–69% and 68–75% (see Table 2) [107]. Cross-reactivity with other fungal infections, including histoplasmosis (similarly to standard GM–EIA testing), has been observed, and is a potential limitation [108]. Anti-mold agents reduce sensitivity [104], thus use in patients on antifungal prophylaxis or treatment may be constrained.

A lateral flow assay using the galactofuranose-specific monoclonal antibody mAb476 was developed for urine POC testing. Sensitivity and specificity were reported at 80% and 92%, with higher sensitivity (90.9%) in hematologic malignancy patients [108,109]. LFDs are inexpensive to produce and can provide rapid easy-to-interpret results without the need for specialized equipment or training. Further clinical studies are needed for broader application.

### 4.3. Aspergillus PCR-Based Testing 

An array of PCR-based assays have been developed for the clinical diagnosis of IA. The updated 2019 Cochrane review including 29 studies of PCR from whole blood, serum, or plasma, showed a pooled sensitivity of 79.2% and specificity of 79.6% for PCR-based testing. For two or more consecutive positive results, sensitivity was lower at 59.6% and specificity improved to 95.1% [110]. Based on a 2012 systematic review, BAL-PCR had a reported sensitivity and specificity of 77% and 94%, respectively [111]. 

The implementation of PCR testing on serum, whole blood, and BAL fluid into clinical practice was previously limited by lack of standardization of techniques, with notable variability in the methods of DNA extraction, primer use, and differences in reference criteria to define a positive result. The European *Aspergillus* PCR Initiative (EAPCRI) group was formed in 2006 to develop methodological guidelines for technique standardization [112]. White et al. 2015 showed that when comparing EAPRCI non-compliant protocols with compliant ones, sensitivity increased from 85% to 98% and specificity from 82% to 87% [112]. Based on the performance of these assays and improved standardization, EORTC/MSG incorporated the use of *Aspergillus* PCR into the diagnosis of probable invasive aspergillosis in September 2020. To meet mycological criteria, patients must have blood (serum, whole blood, or plasma) PCR positivity on two consecutive tests, BAL PCR positivity on two or more tests, or at least one positive test from blood and one from BAL testing [74].

The majority of assays described in the literature were developed in-house, but there are a number of commercial assays now available in a multiplex format that detect *Aspergillus sp.* and resistance mutations [69], including the most prevalent cyp51A gene mutations associated with azole resistance (R34/L98H and TR46/Y121F/T289A mutations) [69,94]. In addition to rapid diagnosis of resistance mutations, PCR amplification allows for the potential to diagnose mixed strains of *Aspergillus* with both azole-susceptible and -resistant isolates that would not be detected by conventional phenotypical susceptibility testing [113]. Expansion of commercial tests to include probes for additional resistance mutations is needed. Roth et al. reviews the diagnostic performance of commercially available *Aspergillus* PCR tests, Including the MycAssay Aspergillus^®^ (Myconostica Ltd., Cambridge UK), AsperGenius^®^ (Pathonostics, Maastricht, The Netherlands), among others (see Table 2) [90].

PCR allows for direct detection of *Aspergillus* DNA in blood, serum, or BAL fluid and has moderate accuracy for screening high risk patients with suspected IA. It has an excellent negative predictive value (~95% with either single or serial testing) and improved positive predictive value with serial performance and/or in combination with other biomarkers [110]. Compared to GM, PCR is more sensitive but slightly less specific, while serial positive PCR is less sensitive but more specific. Unlike GM and BDG which are released during active disease, *Aspergillus* DNA may be detected in the absence of active angio-invasive disease. Though this assay does not distinguish between active disease and colonization, it does provide a potential opportunity for early initiation of either pre-emptive therapy in those with high clinical suspicion but inconsistent radiographic findings, or antifungal prophylaxis in at-risk individuals [110,114]. PCR could be incorporated as part of a screening strategy for ruling out disease, rather than initiating empiric antifungal therapy in high-risk groups [110].

There are still a number of limitations with the use of PCR. The impact of antifungal therapy on test sensitivity is not well defined. False positivity (up to 12% [115]) due to cross reactivity with other mold species or environmental contamination remains a concern. Though the meta-analyses described include a spectrum of patients, the majority of PCR-based studies have been applied to patients with hematologic malignancies, thus limiting some extrapolation to solid organ transplant patients or other hosts where the burden of disease may be less, and assays potentially less sensitive [102].

**Table 2 jof-07-00127-t002:** Commercially available non-culture-based testing for *Aspergillosis* and Mucorales.

Test Name	Example Commercial Product	Sample Source	TAT	Disadvantages	Sensitivity	Specificity	Notes	Citations
1,3-β-D-glucan (BDG)	Fungitell (Associates of Cape Cod, Inc.) and Fungitec G-MK. (Seikagaku).	Serum	Fungitell STAT (qualitative): 40–60 min Regular Fungitell: 24–72 h (d)	Cross-reactive with other fungi, False positives frequent. Often run in reference labs.	Fungitell: 33–100% Fungitec: 67–88%	Fungitell: 36–94% Fungitec: 84–85%	FDA approved.	[116]
Galactomannan	Platelia *Aspergillus* EIA/Ag (Bio-Rad)	Serum, BAL (also CSF, pleural fluid)	1–7 days	Cross-reactive with other fungi. False positives frequent.	**Neutropenic/heme malignancy** Serum: 61–79% BALF: 58–90% **Non-neutropenic:** Serum: 38–41% BALF: 65–76%	**Neutropenic/heme malignancy** Serum: 81–95% BALF: 84–96% **Non-neutropenic:** Serum: 87–89% BALF: 81–90%	FDA approved. Serially monitoring can assess treatment response.	[117,118,119,120,121]
Lateral flow devices	AspLFD (OLM Diagnostics) and the Aspergillus galactomannan LFA (IMMY)	Serum, BAL, urine	15–30 min	Serum LFD requires additional preparation steps/pre-treatment. Sensitivity decreased with antifungals.	**AspLFD:** **Neutropenic/heme malignancy:** Serum: 56–68% BAL: 71–89% **Non-neutropenic:** BAL: 46–69% **LFA:** **Neutropenic/heme malignancy:** 89–97% **Non-neutropenic:** BALF: 65–69%	**AspLFD:** **Neutropenic/heme malignancy:** Serum: 87–90% BAL: 88–100% **Non-neutropenic:** BAL: 46–58% **LFA:** **Neutropenic/heme malignancy:** 88–98% **Non-neutropenic:** BALF: 62–68%	Available in Europe. Urinary GM-like antigen-based test also exists but needs further validation.	[103,104,105,106,107,109]
Aspergillus PCR	MycAssay Aspergillus (real-time PCR) AsperGenius assay (multiplex real-time PCR)	Serum, BAL	12–24 h	Sensitivity decreased by antifungal treatment. Many commercially available assays. Standardization efforts ongoing.	**Serum:** 60–79% **BALF:** 77%	**Serum:** 80–95% **BALF:** 94%	Some detect azole-resistant mutations. Independent validation still needed for most.	[90,110,111,112]
Mucorales PCR	MucorGenius (Pathonostics)	BAL, biopsy fluid	3 h	Small clinical studies.	**90–100%**	**90–99%**		[122,123]

### 4.4. Radiotracers

The diagnosis of IPA requires chest imaging, though abnormalities on basic chest tomograms (CT) are often non-specific and difficult to distinguish from other forms of invasive mold infections. Combining CT and positron emission tomography (PET) with [18F]fluorodeoxyglucose, a marker of metabolic activity, helps to localize an area of abnormality but does not distinguish between malignancy, infection, or inflammation [124]. A number of radiotracers have been developed to better image IPA and could theoretically be useful as adjunctive diagnostic tools to visualize infected tissue and monitor clinical response to treatment [125].

In preclinical murine experiments with two ^99m^Tc labeled *Aspergillus*-specific fungal rRNA-targeted Morpholino oligomers (MORF) probes, researchers observed that probe accumulation is two times higher in infected lungs than non-infected lungs on single-photon emission tomography (SPECT)/CT imaging. One of the probes (AGEN) was *Aspergillus* genus specific but had some cross reactivity with *C. albicans* while the other (AFUM) was species specific for *A. fumigatus* only, limiting its scope of detection [126].

Another imaging technique that has been developed combines microPET/CT with the detection of iron-scavenging siderophores. Siderophores are specific iron-chelating molecules secreted by fungi, which act as virulence factors. *A. fumigatus* and *A. nidulans* produce the siderophores triacetylfusarinine C (TAFC) and ferricrocin (FC), which, when combined with ^68^Ga, a radionuclide with complexing properties similar to that of iron, can be visualized on microPET/CT [127,128]. This method allows for diagnosis, localization, and potentially determination of severity of diseases by degree of uptake, though its application to other species of *Aspergillus* is lacking (e.g., *A. terreus* and *A. niger* produce other siderophore types [129]. Recent studies using fluorescent dyes highlight the potential for hybrid imaging in localizing infection [130]. The effect of antifungal prophylaxis, the degree of iron overload (competing with radiolabeled siderophores), and observed cross reactivity with other species are potential limitations to test sensitivity and specificity [125,127,131]. This modality has also not been well described for the identification of other species of *Aspergillus* aside from *A. fumigatus*. Further studies have evaluated the sensitivity of other siderophores [127] and explore the possible use of other radionuclides with longer half-lives for longitudinal monitoring [132].

The use of imaging combined with antibody detection has been well described in cancer diagnostics. The high specificity of mAbs, as previously mentioned, for diagnosing IA makes antibody-guided imaging techniques an attractive and highly specific way to both detect and visualize IA. In a murine model of neutrophil depleted mice infected with *Aspergillus*, [^64^Cu]DOTA-labeled mAb mJF5, a monoclonal antibody specific to a mannoprotein antigen of *Aspergillus sp.* released during active fungal growth, was effective in localizing an area of the lung infection with PET/MRI while discriminating between active infection and colonization from other pathologies [124]. Given the long half-life of [^64^Cu]DOTA-mJF5, serial imaging could potentially be applied for monitoring response to antifungal treatment and progression of disease [124]. Immuno-PET MR has a broader diagnostic spectrum for *Aspergillus sp.* than the above-described techniques. The JF5 has recently been engineered for use in humans with preclinical studies showing enhanced diagnostic performance compared to its murine counterpart, bringing immuno-PET-MR closer to the clinical landscape [133]. The assessment of semi-invasive aspergillosis syndromes (e.g., in patients with COPD or other chronic lung disease) or the detection of extrapulmonary infections are areas of potential exploration [125].

### 4.5. Volatile Metabolite Profiles of Aspergillus

Another non-invasive method of *Aspergillus* detection utilizes exhaled air for detection of volatile organic compounds (VOCs) released in breath in the setting of IA [134,135]. A proof-of-concept study demonstrated distinct VOC signatures with 100% sensitivity and 83% specificity in high-risk hematologic malignancy patients using “electronic nose” technology [134]. In a prospective study of 64 patients with hematologic malignancies, detection of specific secondary metabolite volatile organic compounds using thermal desorption/gas chromatography/mass spectrometry (α-trans-bergamotene, β-trans-bergamotene, a β-vatirenene-like sesquiterpene, and trans-geranylacetone) had a sensitivity and specificity of 94% and 93%, respectively, for IA [135]. Volatile metabolite profiles could be useful biomarkers for rapid and inexpensive diagnosis of IFI, but are pending further clinical validation. The relationship between metabolite signature and nodule size, the kinetics of these metabolites with antifungal therapy, and distinguishing between colonization versus infection are all areas of potential exploration [135]. This technology could be applied for the detection of other pathogenic molds and endemic fungi [136].

## 5. Mucorales

Invasive mucormycosis, caused by filamentous fungi of the order Mucorales, is the second most common invasive mold infection after invasive aspergillosis [137]. Mucorales causes significant morbidity and mortality in immunocompromised hosts and in patients with poorly controlled diabetes mellitus and ketoacidosis. Diagnosis is usually made by culture and histopathology and is essential to guide mold active therapy, as several first-line antifungal agents lack therapeutic efficacy against Mucorales. Given the low yield of biopsy and culture, patients are often started on empiric therapy for suspected disease with broad spectrum antifungal therapy that covers both *Aspergillus* and Mucorales, as distinguishing between these two entities without definitive confirmation can be challenging. Molecular methods are typically employed for species identification and detection when cultures are negative and can detect potentially mixed infections [138]. There are no commercially available serological tests, though this is an area of active development.

### 5.1. Mucorales-Specific PCR

In addition to PCR from tissue biopsy samples, serum and BAL PCR assays have been used to detect Mucorales from clinical samples. BAL PCR may be a useful adjunctive test to allow for earlier initiation of antifungal therapy and detection in culture-negative BAL samples [139]. One study of BAL fluid from 374 immunosuppressed patients with pneumonia used a combined approach of three qPCR assays on BAL fluid. A total of 24 patients had a positive BALPCR; 23/24 met radiologic criteria for IMI, of which 7 had proven and 3 had probable mucormycosis, 5 had other fungal infections, and 8 had possible IFD. Sensitivity and specificity for probable or biopsy-proven pulmonary mucormycosis was 100% and 97%, respectively. Only 2/24 PCR positive samples had concordant positive cultures [139]. PCR combined with high-resolution melt analysis (PCR/HRMA) has also been described, and showed a sensitivity and specificity of 100% and 93% in one study of 99 BAL samples (9 of which were positive) [122]. MucorGenius (Pathonostics, Maastricht, The Netherlands), is a non-FDA approved semi-quantitative PCR assay that targets 28S rRNA in BAL and biopsy samples and can be run in parallel with AsperGenius, with a TAT of 3 h [123,140].

Non-invasive techniques to detect Mucorales PCR from plasma, serum, or urine are desired to avoid biopsy and even BAL in critically ill patients unable to tolerate these procedures. qPCR using genera-specific, broad-range, or multiplex PCR from serum has been described as successful in detecting infection as early as up to 28 days prior to mycological diagnosis [138,139,141,142] and up to 3 days earlier than classic radiographic findings [143]. Millon et al. reported a sensitivity between 81% and 92% when combining 3 genera-specific real-time qPCR assays, with notably higher sensitivities using larger sample volumes (1 mL) [141]. Sensitivity of PCR is reduced in those receiving antifungal therapy, a notable limitation [141,143]. Persistent DNA detection despite antifungal initiation was associated with higher mortality, suggesting a possible application for serial sampling in prognostication or treatment monitoring [141]. PCR techniques could be considered for screening high risk patients [141] and efforts to standardize PCR techniques will allow for broader application in the future [138].

### 5.2. Other Biomarkers

The detection of mold-reactive CD154+ cells has been suggested as a non-invasive (TAT ~24 h) way to detect invasive Mucorales. *Mucorales*-specific T cells were identified via enzyme-linked immunospot (ELISpot) and were found to be reactive only in patients with proven invasive mucormycosis (IM). These CD4+ or CD8+ cells produced IL-4, IL-10 (Th2)^4^, IFN-γ, and IL-17, but only during the course of active infection, not detected before or after [144,145]. Steinbach et al. quantified mold-reactive CD4/CD69/CD154+ lymphocytes with flow cytometry and found a sensitivity of 100% and a specificity of 81% for Mucorales infection in a cohort of 115 at risk patients (4 with proven, 3 with probable, and 44 with possible IMI), with a TAT of about 24 h. Though only a small number of patients with Mucorales were included, this test could theoretically be used to rule out disease or make an earlier diagnosis prior to disease manifestation/progression. Patients with T cell counts <4500 were excluded, thus limiting extrapolation to those with severe bone marrow suppression and T-cell dysfunction. Given the underlying immune deficiencies in patients at risk for mucormycosis, this is a significant limitation to this approach [146].

Burnham-Marusich et al. developed a pan-fungal monoclonal antibody, 2DA6, that reacts with purified mannans of different fungi, including *Rhizopus*, *Mucor,* and *Aspergillus* and can be detected using enzyme-linked immunosorbent assay (ELISA). A lateral flow format for this test has been developed. Such an assay may be useful in addition to the BDG assay, which is typically negative in *Rhizopus* and *Mucor* infections [147]. Human validation studies are still needed.

Using a method of screening genomic libraries called signal sequence trap by retrovirus-mediated expression (SST-REX), Sato et al. isolated a *Rhizopus-specific* antigen (RSA) and developed a corresponding ELISA assay, which is still pending clinical validation. SST-REX has traditionally been applied for the detection of biomarkers in malignancies, and has the potential for application to fungal diagnostics [148].

## 6. Conclusions

With the growing threat of invasive fungal infections and concurrent rise in antifungal resistance, new technologies have emerged for rapid species identification and earlier detection of IFD. Advancements in proteomics and molecular techniques have allowed for highly discriminatory species identification. Non-culture-based methods including enhanced imaging modalities, T2 magnetic resonance assays, multiplex panels, NGS metagenomic sequencing, volatile metabolites, and new immunologic biomarkers could overcome the prolonged turnaround times and limited sensitivity of traditional techniques, while potentially obviating the need for invasive sampling. Lateral flow devices, microfluidics, and microarrays are promising platforms for clinical integration.

Further studies are needed to define the performance characteristics of many of these technologies and their clinical impact on patient outcomes. When the gold standard is relatively insensitive, defining the precise performance characteristics of non-culture-based techniques is challenging. A rational approach to adjudicating results must be applied where an assay that theoretically outperforms the gold standard produces discordant results. Even highly sensitive and specific tests have limited positive predictive value when applied to patient populations at low risk of IFD, so implementation of new technologies as a part of “screening” algorithms must be performed judiciously. Considerations of capital equipment costs and laboratory staff training must be weighed against the potential cost-saving benefits of earlier diagnosis, given the high costs associated with IFD and IFD-related hospitalizations.

**Table 3 jof-07-00127-t003:** Novel diagnostics for the detection of fungal infections. In development: the test is still undergoing preclinical or clinical validation and has been tested on a limited number of samples. Research only: can be performed in specialized laboratories but is not commercially available or FDA-approved yet.

Diagnostic Test	Target	Stage of Development	Notes	Citations
Monoclonal ab for *Candida*	β-glucan and MP65	Murine models, in development	Not species specific	[34]
Interleukin-17	IL-17 detection	Research	Can help distinguish active infection versus colonization, non-specific	[35]
Immunochromatographic assays for Candida (LFD or DPP)	Antibodies to *C. albicans*	In development, human and laboratory derived samples, no clinical validation	Can be applied to rapid POC testing, false positivity	[42,43]
Microfluidic device to detect *Candida sp.*	*Candida* cells	In development, performed on spiked samples	Cell wall lysis precludes susceptibility testing, can be applied to POC settings	[60]
Next-generation sequencing, e.g., Karius Test (Karius)	Cell-free DNA from yeasts or molds	Research, commercially available, not FDA approved	Further clinical validation studies needed for routine use in fungal diagnostics, limited sensitivity, performed in reference labs, expensive	[83]
Rapid evaporative ionization mass spectrometry (REIMS)	MS analysis based on lipid content of cells (has been applied to *Candida*)	Research, application in development, performed on human samples	Can be coupled with electrosurgery for intraoperative diagnostics	[89]
Raman spectroscopy Surface enhanced RS (SERS), e.g., RenDx Fungiplex (Renshaw Diagnostics) Interference enhanced RS	Fungal PCR from blood/serum samples TAFC fungal siderophores in urine	Research, RenDx Fungiplex is commercially available (clinical utility uncertain), Research, application in development, performed on spiked urine samples	high sensitivity, expensive, can be performed directly on clinical specimens Slightly less sensitive, cheaper, high resolution, rapid TAT <3 h, multiplexing potential	[87,88,91]
PCR coupled with electrospray-ionization mass spectrometry (PCR/ESI-MS)	Species-specific PCR sequences from fungal pathogens (performed on visualized hyphae of Mucorales)	In development	Has only been performed in Mucorales, TAT 6 h Expensive	[89]
Radiotracers for the detection of *Aspergillus*	^99m^Tc labeled *Aspergillus*-specific fungal rRNA-targeted Morpholino oligomers (MORF) probes using SPECT/CT imaging MicroPET/CT to detect TAFC iron scavenging siderophores for detection of *Aspergillus* [^64^Cu]DOTA-labeled mAb mJF5 detects mannoprotein antigen of *Aspergillus*	In development, murine experiments In development In development, murine studies	Limited diagnostic scope, possible cross reactivity with other fungi Limited species detection, anti-mold therapies and iron overload reduce sensitivity Long half-life allows for monitored response to treatment	[124,126,127,128].
Volatile metabolite profiles	Exhaled volatile organic compounds for detection of *Aspergillus* and other molds	In development, murine and human studies	Can be applied to POC testing, breath sampling is non-invasive	[134,135]
Mucorales-specific T cells	Mold-reactive CD154+ cells by ELISpot	In development, human studies	Studies not performed in patients with severe T cell depletion/dysfunction	[146]
Pan-fungal monoclonal antibody (2DA6)	Purified mannans from molds	In development (ELISA and LFD)	Non-specific assay, human validation needed	[147]
*Rhizopus* ELISA	*Rhizopus*-specific antigen (protein RSA)	In development	Detects only *Rhizopus oryzae*	[148]

## Figures and Tables

**Table 1 jof-07-00127-t001:** Commercially available blood culture-independent diagnostic modalities for detecting *Candida sp.* IC: invasive candidiasis. IA: intra-abdominal. DS: deep seated.

Test Name	Example Commercial Products	Sample Source	TAT	Disadvantages	Sensitivity	Specificity	Notes	Citations
1,3-β-D-glucan (BDG)	Fungitell, Fungitell STAT (Associates of Cape Cod, Inc.) and Fungitec G-MK. (Seikagaku). Wako β-glucan (Fujifilm Wako Chemicals)	Serum	Fungitell STAT (qualitative)40–60 min Fungitell: 24–72 h 120 min	Not specific for Candida (e.g., can be + with invasive aspergillosis, fusariosis, *Pneumocystis jirovecii* infection) High false positives Often run in reference labs Lower sensitivity	IC: 75–80% IA/DS: 56–77%	IC: ~80% IA/DS: 57–83%	FDA approved in 2004, better performance with two consecutive results Available in Europe, does not require batch testing	[8,22,23]
Candida mannan	Pastorex Candida (Bio-Rad) Platelia Candida Ag Plus (Bio-Rad)	Serum or plasma	2 h	May form immune complexes and be rapidly cleared	IC: 58%	IC: 93%	Available in Europe	[8]
Combined mannan/antimannan	Platelia Candida Ag-Plus and Ab-Plus (Bio-Rad) Serion Mannan Kit (Serio GmbH)	Serum or plasma	2 ½ h	Low sensitivity due to rapid clearance and complex formation with antibodies	IC: 83% IA/DS: 40%	IC: 86% IA/DS: 25%	Available in Europe	[8,24,25]
T2 Candida nanodiagnostic panel	T2 Candida (T2 Biosystems)	Whole blood	4.4 +/− 1 h	Identifies limited number of Candida species (only 5 most common) High cost. Needs further validation in IA/DS	IC: 91% IA/DS: 33%	IC: 94% IA/DS: 93%	FDA approved	[26,27]
C. albicans germ tube antibody assays (CAGTA)	CAGTA; Vircell Kit and VirClia IgG Monotest	Serum	~3 h	Lower sensitivity for *C. tropicalis*	IC: 42–96% IA/DS: 53–73%	IC: 54–100% IA/DS: 54–80%	Not FDA approved (used in Europe) Increased accuracy when combined with BDG	[28]
Candida PCR performed directly on clinical specimens	LightCycler. SeptiFast (Roche Diagnostics), SepsiTest (Molzym), Magicplex system (Seegene), or VYOO. (SIRS-Lab),	Whole blood, serum, plasma	Minutes to hours (real-time PCR). Multiplex PCR: 4–12 h	Not standardized or validated in multicenter trials. False negatives (low burden of fungal cells in blood, difficulties with sample preparation and DNA extraction) and false positives (similarities with human DNA, sample contamination)	IC: 73–95% IA/DS: 86–91%	IC: 92–95% IA/DS: 33–97%	None FDA approved Variety of DNA targets including Candida-specific genes or broad range pan-fungal genes	[24,29]

## Data Availability

Data sharing not applicable.

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
