# Peer review of "The Evolving Landscape of Fungal Diagnostics, Current and Emerging Microbiological Approaches"

_jof, 2021, doi:10.3390/jof7020127_

Round 1
Reviewer 1 Report
The review by Weiss et al. provides interesting and up-to-date diagnostic information for the detection of Candida and the most common invasive fungal infections, including Aspergillus and Mucorales.
Minor concerns:
- I recommend checking the italic form of the text. There are paragraphs to correct and review
- It would be nice to change the format of the summary Tables. In Table 2, the data on sensitivity and specificity / type of patient are not easy to understand
Author Response
- I recommend checking the italic form of the text. There are paragraphs to correct and review. This formatting issue is now resolved.
- It would be nice to change the format of the summary Tables. In Table 2, the data on sensitivity and specificity / type of patient are not easy to understand. We Added explanations for IC, IA, DS below table title. The format of the summary table was also adjusted.
Reviewer 2 Report
This review does an excellent job of describing and critiquing both commonly used FDA-approved techniques for fungal identification as well as new techniques in development. This will be useful for medical practitioners, clinical diagnostic laboratory practitioners and those interested in identifying opportunities for new assay development. It is well-written and complete. There are only a few issues that are noted and suggestions are made to make the review more accessible and impactful.
Issues:
- Within the text, it is not always clear which assays are in development, commercialized for research and FDA-approved. These types of assays should be flagged by class when they are first mentioned. Since tests in development are described, it would be useful to have a secondary description of those tests in another table. In addition, it would be useful to add a table to supplement Tables 1 and 2, that would briefly list all of the assays based on their stage of development, including all of the assays that are discussed.
- A clear sidebar with acronyms would be very useful, or perhaps several sidebars throughout the review paper
- While the focus on candidemia is warranted, a brief discussion of other diseases caused by Candida would be useful, along with comments on diagnosis of these diseases.
- It seems unlikely that the TAT for NGS sequencing is 12-24 hr, when it has to be sent out to a reference lab.
- Issues with italics several places (lines 417-438, 456-467).
- (of aspergillus?) on line 508 should be resolved.
- Lines 723-725. Description of CotH assay is confusing. Is this PCR, like everything else in the paragraph? If so, this sentence should describe the CotH gene rather than the protein.
- Lines 741-742. Limitation of the study is noted that immunosuppressed individuals may be confounding for CD154+ assays. Was this group included in the study cited? If so, then the study showed that the assay can work in this group. If not, this should be clarified.
- Lines 779-780. Please expand the description of COIs. Which assays described in the text are relevant for these COIs? Please include this information here in the COI section.
- Tables 1 and 2 might be best as landscape. Lack of formatting is distracting.
Author Response
- Within the text, it is not always clear which assays are in development, commercialized for research and FDA-approved. These types of assays should be flagged by class when they are first mentioned. Since tests in development are described, it would be useful to have a secondary description of those tests in another table. In addition, it would be useful to add a table to supplement Tables 1 and 2, that would briefly list all of the assays based on their stage of development, including all of the assays that are discussed. We have included an additional table which outlines the assays that are currently either in research only or development stages. Throughout the text we have made it more clear as to whether assays were available commercially or in development or just used in research settings
- A clear sidebar with acronyms would be very useful, or perhaps several sidebars throughout the review paper- This has been added but can be broken into multiple sections if needed
- While the focus on candidemia is warranted, a brief discussion of other diseases caused by Candida would be useful, along with comments on diagnosis of these diseases. In the first paragraph we have expanded on the clinical presentation of candida infection to capture its spectrum of disease. We have adjusted the text to include more information about the diagnosis of hepatosplenic candidiasis. We also include mention of lateral flow assays for candida vaginitis. Since the focus of this paper was on diagnostics for invasive candidiasis, we did not include extensive discussion of diagnosis in local infection.
- It seems unlikely that the TAT for NGS sequencing is 12-24 hr, when it has to be sent out to a reference lab. Text was adjusted to clarify TAT is based on when it is received by reference labs
- Issues with italics several places (lines 417-438, 456-467). Resolved
- (of aspergillus?) on line 508 should be resolved.- resolved
- Lines 723-725. Description of CotH assay is confusing. Is this PCR, like everything else in the paragraph? If so, this sentence should describe the CotH gene rather than the protein. Replaced “spore coating protein (CotH)” for “CotH genes”, this section was subsequently removed for brevity
- Lines 741-742. Limitation of the study is noted that immunosuppressed individuals may be confounding for CD154+ assays. Was this group included in the study cited? If so, then the study showed that the assay can work in this group. If not, this should be clarified. Patients with T cell counts below the threshold (4500) were excluded from the analysis, so its use for those with severe bone marrow suppression and T-cell dysfunction may be challenging, which is a significant limitation of this approach- the text was adjusted to reflect this,
- Lines 779-780. Please expand the description of COIs. Which assays described in the text are relevant for these COIs? Please include this information here in the COI section. Dr Koo is involved with Fujifilm diagnostics (we discuss the WAKO diagnostic), the other conflicts don’t specifically make fungal diagnostics however most publishers require all COI listed so we included all, the others. (Merck) may be removed if felt not to be important by publishers
- Tables 1 and 2 might be best as landscape. Lack of formatting is distracting. This is resolved in the new version
Reviewer 3 Report
This paper by Weiss et al is an extensive overview of current and future diagnostic methods for invasive fungal infections. It covers a very wide array of diagnostic tools for an array of the most clinically relevant causative agents. This broad spectrum of course means that not every method is described in full detail, but I feel the authors found a good balance between the amount of detail and the different topics that are covered. This paper could be a great reference work for anyone starting out in the field of fungal diagnostics today.
I only have some minor questions, suggestions and remarks:
- Lines 82-85: MALDI-TOF susceptibility testing appears promising, but other studies have found that the agreement with conventional testing is not 100% (eg DOI 10.3389/fmicb.2019.03000). It therefore appears useful as a fast screening assay, but conventional testing is still required
- Lines 91-94: The authors mention MALDI-TOF as a useful test for determining the strain of C. auris. Maybe they can add that MALDI-TOF has been shown to be more reliable in detecting and correctly identifying C. auris than conventional techniques that often mislabel C. auris (depening of course on the appropriate MALDI-TOF library)
- Lines 112-113: Wako FUJIFILM has recently obtained a CE mark in Europe for IVD use, and overcomes a lot of the disadvantages of the Fungitell assay (significantly less hands on time, random access, less variability). Several comparative studies have already been published
- Line 141: remove the duplicate word "these"
- Line 147: the word "been" is missing
- Lines 152-163: the patients at highest risk for IFI and with the highest frequency of testing for IFI typically are severely immunocompromised, making these tests only useful in niche patient groups or in research settings for drug/vaccine development
- Line 207: please define abbreviation TAT upon first use
- Line 305: should this be PNA-FISH?
- Lines 346-350: a closing bracket is missing. Furthermore, this seems to be referring to the EORTC-MSGERC criteria which are meant for research use, not for clinical use, which can include patients with a much broader clinical presentation regarding risk factors (influenza, COPD, cirrosis, ...) and radiologic abnormalities.
- Paragraph 3.2: something appears to have gone awry with the italics in this paragraph. Furthermore, this paragraph contains a lot of very detailed information, especially compared to the amount of detail in other paragraphs. This makes it very difficult to follow and to read. Please summarize in a similar fashion as the other paragraphs.
- Line 407: a word seems to be missing at the end (packages?)
- Lines 462-465: this seems to be a contradiction. If NGS has a low sensitivity, why would you use it as an adjunct when other tests are already negative? From the preceding lines, I would infer the sensitivity to be good (though not perfect), rather than low?
- Line 499: the word "in" is missing
- Lines 495-500: this paragraph only mentions Candida and no moulds, and therefore appears to be in the wrong section. Move to section on Candida?
- Line 508: please remove brackets and question mark
- Line 509: TAFC is highly specific (only A. fumigatus, A. nidulans and Fusarium graminearum) which is a disadvantage as mentioned by the authors for other techniques
- Line 530: in my experience, serum testing is mostly done in neutropenic/hematology patients. Most other patient populations depend on BAL testing, which is not mentioned in this paragraph despite having a better sensitivity (in case of GM).
- Lines 539-541: please specify that this is for serum GM, not BAL GM
- Lines 542-543: please add a reference for the use in the monitoring of treatment response as this use is not always clear for physicians with a low incidence of IA (when to retest, values to target, ...)
- Lines 548-550: assigning a TAT of several days to GM testing seems unfair in comparison to other tests discussed in this paper. The GM assay itself is a simple sandwich ELISA, and thus has a TAT of a couple of hours if performed routinely. In reality, this is often indeed several days due to batching, but this happens with others tests such as PCR as well.
- Lines 549-550: these alternative assays currently still lack high quality validation
- Line 568: as the authors mention elsewhere in this paper, NPV/PPV depend on the prevalence. It is therefore not very informative to provide these values from a "mixed" population without also providing the exact composition of this population (ICU? Hematology? SOT?) or the prevalence of disease. Specificity and sensitivity seem more appropriate here, although the population being tested can influence this as well of course
- Lines 571-572: I disagree, as administration of mould-active therapy >24h before testing has been shown to result in a significant reduction in sensitivity (eg DOI 10.1093/mmy/myz079 or others)
- Line 613: replace the word PCR by DNA
- Line 617: this may be a semantic discussion, but is it still considered prophylaxis if you have already detected the presence of the pathogen?
- Section 5.1: in the section on Aspergillus-PCR, the authors mention and reference several commercial PCR assays, which is indeed very relevant for labs without in-house developed assays. However, similar commercially available assays seem to be missing in this section (eg MucorGenius)?
Author Response
This paper by Weiss et al is an extensive overview of current and future diagnostic methods for invasive fungal infections. It covers a very wide array of diagnostic tools for an array of the most clinically relevant causative agents. This broad spectrum of course means that not every method is described in full detail, but I feel the authors found a good balance between the amount of detail and the different topics that are covered. This paper could be a great reference work for anyone starting out in the field of fungal diagnostics today.
I only have some minor questions, suggestions and remarks:
- Lines 82-85: MALDI-TOF susceptibility testing appears promising, but other studies have found that the agreement with conventional testing is not 100% (eg DOI 10.3389/fmicb.2019.03000). It therefore appears useful as a fast screening assay, but conventional testing is still required. Adjusted text to reflect this “MALDI-TOF susceptibility testing appears promising, however agreement between conventional testing and MALDI may vary, thus conventional testing is still required”
- Lines 91-94: The authors mention MALDI-TOF as a useful test for determining the strain of C. auris. Maybe they can add that MALDI-TOF has been shown to be more reliable in detecting and correctly identifying C. auris than conventional techniques that often mislabel C. auris (depening of course on the appropriate MALDI-TOF library). Text added.” MALDI-TOF MS has been shown to be more reliable in correctly identifying species of Candida (such as C. auris) than conventional techniques that may be prone to mislabeling closely related isolates, though this is dependent on the available spectra libraries[15]”
- Lines 112-113: Wako FUJIFILM has recently obtained a CE mark in Europe for IVD use, and overcomes a lot of the disadvantages of the Fungitell assay (significantly less hands on time, random access, less variability). Several comparative studies have already been published. This assay was added to the text and table
- Line 141: remove the duplicate word "these" Resolved
- Line 147: the word "been" is missing Resolved
- Lines 152-163: the patients at highest risk for IFI and with the highest frequency of testing for IFI typically are severely immunocompromised, making these tests only useful in niche patient groups or in research settings for drug/vaccine development text added“Given the profound immunologic derangements in patients at highest risk for IFI, these techniques may be best applied in research settings, drug, and vaccine development. “
- Line 207: please define abbreviation TAT upon first use Resolved
- Line 305: should this be PNA-FISH? Resolved
- Lines 346-350: a closing bracket is missing. Resolved
- Furthermore, this seems to be referring to the EORTC-MSGERC criteria which are meant for research use, not for clinical use, which can include patients with a much broader clinical presentation regarding risk factors (influenza, COPD, cirrhosis, ...) and radiologic abnormalities. The text was adjusted to explain that clinical judgment typically does include host consideration, radiographic criteria, and mycologic evidence. We did not include all host risk factors, just sample ones (eg, SOT, HCT…) but we did add the lines “ The EORTC-MSGERC criteria outline specific guidelines for categorizing IMI into possible, probable, or proven disease, however the classification system is designed for research purposes rather than directing clinical management” to clarify that while clinical judgment includes those criteria, there are research criteria designed to address this with pre-specified guideliens
- Paragraph 3.2: something appears to have gone awry with the italics in this paragraph. Furthermore, this paragraph contains a lot of very detailed information, especially compared to the amount of detail in other paragraphs. This makes it very difficult to follow and to read. Please summarize in a similar fashion as the other paragraphs. These paragraphs have been reduced and summarized
- Line 407: a word seems to be missing at the end (packages?) adjusted
- Lines 462-465: this seems to be a contradiction. If NGS has a low sensitivity, why would you use it as an adjunct when other tests are already negative? From the preceding lines, I would infer the sensitivity to be good (though not perfect), rather than low? Adjusted the text to read “NGS may be useful as an adjunctive test in cases where invasive biopsy is contraindicated”- ---to commenter, its use when other tests are negative is not clearly defined but is an area of active research”
- Line 499: the word "in" is missing Resolved
- Lines 495-500: this paragraph only mentions Candida and no moulds, and therefore appears to be in the wrong section. Move to section on Candida? At the end of the MALDI-TOF section for candida we included in the text “ Other promising spectroscopy techniques have been proven experimentally to detect Candida, though have yet to be applied in clinical laboratories and will be combined in a later section on spectroscopy techniques applied to yeasts and/or molds.” The wording was adjusted to make this more clear (eg to indicate to the reader that we will discuss both at a later time), however we included REIMS in this section to keep the section on additional spectroscopy techniques as concise as possible and in one location. If this is still confusing or out of place, we can split the topic.
- Line 508: please remove brackets and question mark Resolved
- Line 509: TAFC is highly specific (only A. fumigatus, A. nidulans and Fusarium graminearum) which is a disadvantage as mentioned by the authors for other techniques. We included a remark to see later section 4.4 wher TAFC is discussed and we report the limited scope of TAFC for detecting other species of aspergillus
- Line 530: in my experience, serum testing is mostly done in neutropenic/hematology patients. Most other patient populations depend on BAL testing, which is not mentioned in this paragraph despite having a better sensitivity (in case of GM). We include reference to table 2 where sensitivity between BAL GM and serum is outlined
- Lines 539-541: please specify that this is for serum GM, not BAL GM. “in serum samples” added to text.
- Lines 542-543: please add a reference for the use in the monitoring of treatment response as this use is not always clear for physicians with a low incidence of IA (when to retest, values to target, ...) “ though outside the scope of this discussion (re serial monitoring) in the text, we included the two following references for monitoring treatment response
- Kovanda LL, Kolamunnage-Dona R, Neely M, Maertens J, Lee M, Hope WW. Pharmacodynamics of Isavuconazole for Invasive Mold Disease: Role of Galactomannan for Real-Time Monitoring of Therapeutic Response [published correction appears in Clin Infect Dis. 2017 Oct 15;65(8):1431-1433]. Clin Infect Dis. 2017;64(11):1557-1563. doi:10.1093/cid/cix198
- Kovanda, L.L., Desai, A.V. & Hope, W.W. Prognostic value of galactomannan: current evidence for monitoring response to antifungal therapy in patients with invasive aspergillosis. J Pharmacokinet Pharmacodyn 44, 143–151 (2017). https://doi.org/10.1007/s10928-017-9509-1
- Lines 548-550: assigning a TAT of several days to GM testing seems unfair in comparison to other tests discussed in this paper. The GM assay itself is a simple sandwich ELISA, and thus has a TAT of a couple of hours if performed routinely. In reality, this is often indeed several days due to batching, but this happens with others tests such as PCR as well. - the text was changed to say Though this can be performed in hours, batch testing and use of reference laboratories can delay TAT to days
- Lines 549-550: these alternative assays currently still lack high quality validation. Text adjusted to read “ There are a number of alternative biomarker detection kits that have come onto the market but still need proper validation”. Added to text.
- Line 568: as the authors mention elsewhere in this paper, NPV/PPV depend on the prevalence. It is therefore not very informative to provide these values from a "mixed" population without also providing the exact composition of this population (ICU? Hematology? SOT?) or the prevalence of disease. Specificity and sensitivity seem more appropriate here, although the population being tested can influence this as well of course. We removed this sentence as we agree that it does not improve clarity
- Lines 571-572: I disagree, as administration of mould-active therapy >24h before testing has been shown to result in a significant reduction in sensitivity (eg DOI 10.1093/mmy/myz079 or others) We adjusted the text to reflect that antimold agents reduce sensitivity, though these studies do not necessarily distinguish between prophylaxis and treatment, either way it makes sense to include the reduced sensitivity with anti-mold agents
- Line 613: replace the word PCR by DNA Resolved
- Line 617: this may be a semantic discussion, but is it still considered prophylaxis if you have already detected the presence of the pathogen? Some sources consider treatment of colonization as prophylaxis (example: intranasal mupirocin for S. aureus: Kevin B. Laupland, John M. Conly, Treatment of Staphylococcus aureus Colonization and Prophylaxis for Infection with Topical Intranasal Mupirocin: An Evidence-Based Review, Clinical Infectious Diseases, Volume 37, Issue 7, 1 October 2003, Pages 933–938, https://doi-org.ezp-prod1.hul.harvard.edu/10.1086/377735) if this remains confusing or should be changed, we can adjust the wording however we did not want to imply that “treatment” is necessary for a positive PCR
- Section 5.1: in the section on Aspergillus-PCR, the authors mention and reference several commercial PCR assays, which is indeed very relevant for labs without in-house developed assays. However, similar commercially available assays seem to be missing in this section (eg MucorGenius) There are many commercially available aspergillus PCR testing We included mention of commercially available mucorales assays
- Guegan H, Iriart X, Bougnoux ME, Berry A, Robert-Gangneux F, Gangneux JP. Evaluation of MucorGenius® mucorales PCR assay for the diagnosis of pulmonary mucormycosis. J Infect. 2020 Aug;81(2):311-317. doi: 10.1016/j.jinf.2020.05.051. Epub 2020 May 28. PMID: 32474046.
- Mercier T, Reynders M, Beuselinck K, Guldentops E, Maertens J, Lagrou K. Serial Detection of Circulating Mucorales DNA in Invasive Mucormycosis: A Retrospective Multicenter Evaluation. J Fungi (Basel). 2019;5(4):113. Published 2019 Dec 3. doi:10.3390/jof5040113
Reviewer 4 Report
The manuscript by Weiss et al provides a very thorough and detailed review of the current state of fungal diagnostics. I tried very hard to find constructive critique, but the authors have done a great job at providing up to date data, written elegantly. Here are a few comments:
Major comments:
- The manuscript is very lengthy. In addition to discussing the more common diagnostic applications, the authors have spent a considerable part of the manuscript discussing proof of concept and research studies – many of which are yet to show clinical performance data. While these are interesting, perhaps the proof of concept technologies can be consolidated. Currently, each research technology has its own paragraph and this can become lengthy (and tiresome) to read if clinical relevance has not yet been demonstrated.
- Tables 1 and 2 are not referenced at all in text. Please add to text.
- Line 478: performance of mold MALDI is also dependent on the test challenge set. Please add to text.
Minor comments:
- Reference 7 has a display/formatting problem that needs to be addressed.
- Phrasing in lines 147 and 499 needs to be addressed.
- Line 191 needs a reference.
- Lines 416 to 467 were italized. Is this a formatting issue?
- Line 449 has a track change that needs to be addressed.
- Line 607. Capitalize Aspergillus.
Author Response
Major comments:
- The manuscript is very lengthy. In addition to discussing the more common diagnostic applications, the authors have spent a considerable part of the manuscript discussing proof of concept and research studies – many of which are yet to show clinical performance data. While these are interesting, perhaps the proof of concept technologies can be consolidated. Currently, each research technology has its own paragraph and this can become lengthy (and tiresome) to read if clinical relevance has not yet been demonstrated. We have consolidated some areas including the topic of NGS, removed some diagnostic tests that were of historical interest but no longer used (d arabinitol) and removed text where it felt redundant. The purpose of this paper was to explore emerging diagnostics so detail on proof of concept studies is felt to be necessary, but we have adjusted text for conciseness where possible.
- Tables 1 and 2 are not referenced at all in text. Please add to text. Adjusted
- Line 478: performance of mold MALDI is also dependent on the test challenge set. Please add to text.- text added
Minor comments:
- Reference 7 has a display/formatting problem that needs to be addressed. – this is the standard format for when the authors are the same as the authors listed in the citation above and was auto-formatted by the reference manager as dictated by the journal
- Phrasing in lines 147 and 499 needs to be addressed. Adjusted line 147 and 499
- Line 191 needs a reference. Addressed, location of correct citation was not obvious
- Lines 416 to 467 were italized. Is this a formatting issue? Unclear why this happened when the article was uploaded but should be fixed now
- Line 449 has a track change that needs to be addressed.- addressed
- Line 607. Capitalize Aspergillus.- resolved